# A small electron donor in cobalt complex electrolyte significantly improves efficiency in dye-sensitized solar cells

Yan Hao[1,*], Wenxing Yang[1,*], Lei Zhang[1], Roger Jiang[1], Edgar Mijangos[1], Yasemin Saygili[2], Leif Hammarström[1], Anders Hagfeldt[1,2] & Gerrit Boschloo[1]

Photoelectrochemical approach to solar energy conversion demands a kinetic optimization of various light-induced electron transfer processes. Of great importance are the redox mediator systems accomplishing the electron transfer processes at the semiconductor/electrolyte interface, therefore affecting profoundly the performance of various photoelectrochemical cells. Here, we develop a strategy—by addition of a small organic electron donor, tris(4-methoxyphenyl)amine, into state-of-art cobalt tris(bipyridine) redox electrolyte—to significantly improve the efficiency of dye-sensitized solar cells. The developed solar cells exhibit efficiency of 11.7 and 10.5%, at 0.46 and one-sun illumination, respectively, corresponding to a 26% efficiency improvement compared with the standard electrolyte. Preliminary stability tests showed the solar cell retained 90% of its initial efficiency after 250 h continuous one-sun light soaking. Detailed mechanistic studies reveal the crucial role of the electron transfer cascade processes within the new redox system.

[1] Department of Chemistry-Ångström, Centre of Molecular Devices, Uppsala University, Box 523, SE-75120 Uppsala, Sweden. [2] Laboratory of Photomolecular Science, Institute of Chemical Sciences and Engineering, École Polytechnique Fédérale de Lausanne, EPFL SB ISIC LSPM, Chemin des Alambics, Station 6, CH-1015 Lausanne, Switzerland. * These authors contributed equally to this work. Correspondence and requests for materials should be addressed to G.B. (email: gerrit.boschloo@kemi.uu.se).

Dye-sensitized solar cells (DSSCs) have attracted much interest in the photovoltaic research field since the beginning of 1990s (ref. 1). DSSCs are attractive because of their high light-to-electricity conversion efficiency, flexibility in terms of colours and appearance, their relatively easy fabrication procedures and low production cost. The redox species, an essential component in DSSCs, affect significantly the charge transfer steps and therefore markedly influence the performance of these devices. In principle, they are required to ensure an efficient regeneration of oxidized dyes, diffuse rapidly in the electrolytic solution and possess fast charge transfer kinetics at the counter electrode. Meanwhile, unbeneficial charge recombination processes should be minimized.

The iodide/triiodide ($I^-/I_3^-$) couple is the most favoured redox couple in the first 20 years' development of DSSCs electrolytes, yielding unrivalled energy conversion efficiencies up to 11.7 and 10.3% for ruthenium-based sensitizers and metal-free organic sensitizers, respectively[2,3]. Despite its good performance, the $I^-/I_3^-$ couple does have some limitations that derive from its corrosive nature and the substantial thermodynamic loss in the dye regeneration process[4,5]. Even though considerable efforts have been done on alternative redox couples, the efficiency of DSSC with new redox systems fell far behind the traditional iodine system before 2010 (refs 6–9).

A breakthrough was achieved in 2010 when Feldt et al.[10] reported the usage of a cobalt polypyridine redox mediator in combination with a blocking organic dye, D35, achieving an impressive efficiency of 6.7%. This result stimulated rapid further exploration of cobalt-electrolyte-based DSSCs. With the advance of new efficient and elegant dyes, DSSCs achieved progressive improvement in the device performance, 12.3% in 2011 (refs 11–14), 13% in 2014 (ref. 15) and recently up to 14.3% in 2015 (refs 16–20). However, the performance of cobalt redox mediators in DSSCs is still limited by rapid electron recombination to the electrolyte, which usually happens for one-electron redox mediators, and furthermore by slow dye regeneration and mass transport problems[10]. Feldt et al.[21] studied the regeneration and recombination kinetics in DSCs using a series of cobalt redox couples and suggested that a driving force of at least 0.4 eV is required in the dye regeneration process using a cobalt complex electrolyte. In other words, a significant thermodynamic loss still exists in the dye regeneration process of cobalt-based DSSCs. Further efforts are needed to explore other redox species that minimize this energy loss; however, it is likely that such redox species also lead to increased recombination. The challenge for further development of DSSCs relies therefore strongly on the possibility of bringing the driving force for dye regeneration below 0.4 eV, while maintaining both fast regeneration and slow recombination.

One solution for this is to introduce an intermediate redox species, which has more positive redox potential than the cobalt complex species, but achieves faster regeneration. For example Cong et al.[22] reported the tandem TEMPO-Co(bpy)3 redox system, in which much faster regeneration dynamics was observed and the faster recombination problem of the TEMPO/TEMPO·+ redox couple is also resolved. Unfortunately, the TEMPO-based DSSCs have a short lifetime, because of the instability of the TEMPO·+ species in the electrolyte under illumination[23]. Here we introduce tris(p-anisyl)amine (TPAA)) as an intermediate redox species in cobalt complex electrolyte. TPAA is a small molecule that possesses reversible electrochemical properties, and is easily synthesized. Previously, it has been applied in solid-state DSSCs as a hole conductor[24,25]. That study demonstrated that TPAA can regenerate the dye rapidly, inspiring us that it could be a potential alternative as

co-mediator, helping to overcome constraints of cobalt-based electrolytes.

Here we demonstrate that TPAA and cobalt complexes form an excellent redox tandem in the electrolyte, leading to the ultrafast dye regeneration kinetics, and slowed down recombination between the electrons in $TiO_2$, and both oxidized dye and oxidized redox species. The introduction of TPAA in the electrolyte resulted in an electron transfer cascade process, which significantly optimizes the overall electron transfer kinetics at the photoanode side. The resulting DSSC devices exhibited remarkable efficiencies, 9.1% for LEG4 dye and 10.5% for the co-sensitized system at $100\,mW\,cm^{-2}$ simulated AM 1.5 conditions, corresponding to a relative improvement of 25% in efficiency compared with the standard cobalt complex electrolyte based DSSCs. Especially, considering the indoor application of DSSCs, it is even more promising that 11.7% of efficiency at 0.46 sun illumination was achieved for co-sensitized DSSCs. Preliminary stability tests showed excellent cell stability with 90% of its initial efficiency remaining after 250 h continuous one-sun light soaking tracked at the maximum power point (MPP). Our findings provide a suitable and easy way forward to more efficient DSSCs.

## Results

**Characterization of TPAA.** The chemical structures of TPAA and $[Co(bpy)_3]^{2+/3+}$ used in the present study are shown in Fig. 1a. Cyclic voltammogram of the TPAA in acetonitrile (ACN) shows two reversible waves (Fig. 1b). The first wave is attributed to the oxidization of TPAA to TPAA·+ (ref. 26), for which a formal reduction potential of 0.79 V versus the normal hydrogen electrode (NHE) is determined. The diffusion coefficient of TPAA was evaluated to be $1.5 \times 10^{-5}\,cm^2\,s^{-1}$ by performing the cyclic voltammetry at different scan rates (Supplementary Fig. 1a,b), ∼1.6 times higher than that of $Co(bpy)_3(PF_6)_2$ reported in literature[10], which ensures an efficient electrolytic diffusion within the electrolyte. Spectroelectrochemical measurements using chronopotentiometry at 0.94 V versus NHE (Fig. 1c,d) demonstrate that TPAA·+ exhibits a strong absorption peak at ∼720 nm in the visible spectrum in ACN, in good accordance with previous studies[27,28].

Supplementary Fig. 2a shows ultraviolet–visible absorption spectrum of TPAA and TPAA·+, Supplementary Fig. 2b shows the absorbance changes of TPAA·+ with titration of $[Co(bpy)_3]^{2+}$ by ∼0.2 equiv. with each addition. The intensity of TPAA·+ absorption peak was clearly reduced by the addition of $[Co(bpy)_3]^{2+}$, which confirms that the redox reaction below occurs, in accordance with thermodynamic considerations based on their redox potentials ($E^{0'}$ $[Co(bpy)_3]^{2/3+}$) = 0.56 V versus NHE[10].

$$TPAA^{\cdot+} + Co(bpy)_3^{2+} \rightarrow TPAA + Co(bpy)_3^{3+}$$

**Photovoltaic performance and opto-electric characterization.** Figure 2a shows the current density–voltage (J–V) characteristics of the DSSC devices fabricated with an organic dye LEG4 as the sensitizer[29] and a standard cobalt electrolyte or TPAA/Co electrolyte. The components of the standard electrolyte are 0.1 M $LiClO_4$, 0.22 M $Co(bpy)_3(PF_6)_2$, 0.05 M $Co(bpy)_3(PF_6)_3$ and 0.2 M 4-tert butylpyridine in ACN, with 0.1 M TPAA was added into the standard electrolyte to form TPAA/Co electrolyte. The detailed photovoltaic parameters are summarized in Table 1. A significant improvement of the DSSCs performance is observed on addition of TPAA to the electrolyte. Specifically, the open circuit potential ($V_{OC}$) increased markedly from 835 to 915 mV with the addition of TPAA (0.1 M) to the electrolyte. The dark

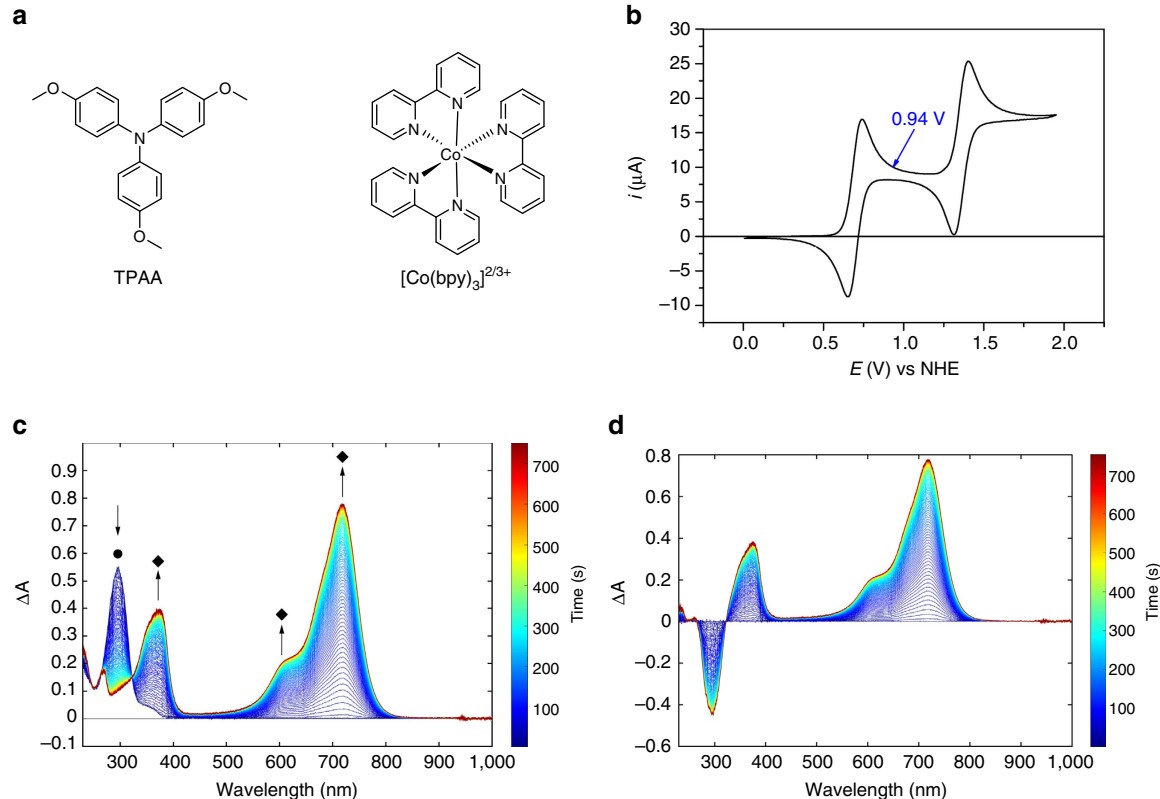

**Figure 1 | Chemical compounds and electrochemistry.** (**a**) Chemical structures of TPAA, $[Co(bpy)_3](PF_6)_2$ and $[Co(bpy)_3](PF_6)_3$ used in the present study. (**b**) Cyclic voltammogram of 1 mM TPAA in 0.1 M TBAPF$_6$ in ACN. The arrow indicates the position of applied potential in the spectroelectrochemical measurements (*vide infra*); working electrode: platinum; counter electrode: graphite rod; reference electrode: Ag/AgNO$_3$ (10 mM in ACN). Conversion to the NHE scale was conducted by calibrating the potential of the reference electrode with Fc/Fc$^+$ redox couple (0.635 V versus NHE) after the measurements (**c**) the evolution of the ultraviolet–visible spectra of TPAA after applying a fixed potential at 0.94 V (versus NHE). Black circle for the absorption peak of TPAA; black diamond marks the absorption peak for TPAA$^{\cdot+}$. (**d**) The graph shows the difference of spectra.

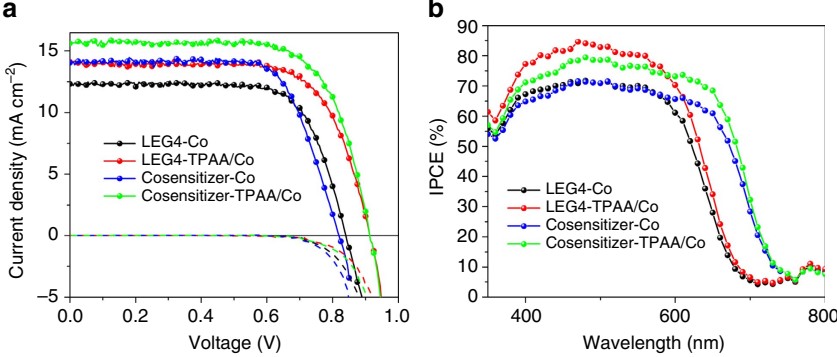

**Figure 2 | JV curves and IPCE.** (**a**) Current density versus applied potential curves under 100 mW cm$^{-2}$ AM 1.5 G illumination (dot lines) and in darkness (dashed lines). (**b**) Spectra of incident photon-to-current conversion efficiency (IPCE) for DSSCs based on LEG4 dye and co-sensitized (D35/Dyenamo Blue).

current of the DSSCs was not significantly altered. The short-circuit photocurrent density ($J_{SC}$) increased from 12.1 to 14.0 mA cm$^{-2}$. This was consistent with the recorded incident photon-to-current conversion efficiency (IPCE) spectra (Fig. 2b) and integral currents from IPCE spectra (shown in Supplementary Fig. 5), showing an increase of the IPCE maxima from 70 to 85%. As a result, the overall efficiency ($\eta$) of the solar cells is boosted from 7.2 to 9.1%.

LEG4-sensitized TiO$_2$ films show an absorption band edge at ~700 nm, which limits the achievable $J_{SC}$ in the present system

due to the weak harvesting of photons in the red part of the solar spectrum. To overcome this, we further utilized a recently developed co-sensitized system (D35: Dyenamo blue = 4:3) as the sensitized films, which effectively broadens the absorption spectrum and has an absorption edge at ~760 nm (ref. 30). As expected, the photocurrent of the co-sensitized system increased significantly (Fig. 2; Table 1). The efficiency of DSSCs with the standard electrolyte was increased to 8.5%, while the DSSCs with the addition of TPAA showed efficiency as high as 10.7%, which corresponds to a >25% improvement of the

efficiency. Light intensity dependence studies showed that the power conversion efficiency of the solar cell peaked at 11.7% under lower light conditions, which further demonstrates the potential of the novel tandem electrolyte, and also suggests suitability for indoor application. In supporting information statistics of a large number of devices are shown in Supplementary Fig. 6, which further clearly demonstrate a 2% absolute increase of $\eta$ by TPAA addition to the electrolyte. It should be noted that the investigated devices did not have an anti-reflecting coating. Further optimization of mesoporous TiO$_2$, dye layer and electrolyte layer will likely increase $\eta$ of this system even further.

To cast insights on the kinetics and energetics of the DSSC, further opto-electrical characterizations based on light intensity modulation techniques were used for characterization[31]. Figure 3a shows charge extraction measurement of solar cells with the two electrolyte compositions. This method yields the accumulated charge ($Q_{OC}$) in the DSSC as function of $V_{OC}$, and gives information relative shift of energy levels. The redox potential of the electrolyte is not affected by TPAA addition and is can be calculated with the Nernst equation, $E = 0.52\,V$ versus NHE (Supplementary Table 1). The good alignment of the extracted charge curves indicates that the energetic band position of TiO$_2$ is not significantly affected by the presence of TPAA. As could be expected, the trap distribution in mesoporous TiO$_2$ is unaffected by the uncharged TPAA molecules.

Figure 3b shows the electron lifetime of the solar cells, characterized by following the voltage decay traces of the DSSCs in response to a small light intensity modulation. The electron lifetimes of all the solar cells were found to decrease exponentially versus $V_{OC}$. This dependence of electron lifetime on $V_{OC}$ is usually explained by the multiple-trapping model in DSSCs. It is clearly shown that the electron lifetime at the same voltage increases significantly with the addition of TPAA, by more than seven times. This corresponds to a largely reduced recombination process in DSSCs, in agreement with the previously improvement of $V_{OC}$ and the decrease of the dark current (Fig. 2). The result is surprising since the Co(bpy)$_3^{3+}$ concentration should be the same in both type of devices.

For completeness, we also tested devices with TPAA/TPAA$^{\cdot+}$ as the redox couple in the electrolyte. These solar cells worked, but had rather poor performance with a $\eta$ of ~2%, see Supplementary Fig. 7 in supporting information. The relatively poor performance is attributed to the rapid electron recombination from TiO$_2$ to the TPAA$^{\cdot+}$ radical cation.

**Charge transfer mechanisms**. We have shown so far that a large improvement of DSSCs performance could be achieved simply by the addition of TPAA in the cobalt electrolyte, and attributed this improvement mostly to an increased $V_{OC}$ and $J_{SC}$ by enhanced regeneration efficiency and diminished recombination kinetics. Considering this simple strategy and significant improvement it brings, the crucial and intriguing questions are that how the addition of TPAA could affect the charge transfer processes of DSSCs in such a marked way. In the following section, the electron transfer processes were investigated with various spectroscopies with a time resolution spanning from femto to millisecond.

Shown in Fig. 4a are the photo-induced absorption (PIA) spectra of LEG4-sensitized solar cells fabricated with inert, TPAA, Co and TPAA/Co electrolyte. The difference spectra in PIA measured at low modulation frequency correspond to the absorption difference of the samples between the ground state and the steady state after the square-wave light excitation. Two characteristic peaks were found in the PIA spectrum of cells with inert electrolyte at ~670 and 770 nm, which has been previously demonstrated to be the characteristic absorption peaks of the oxidized LEG4$^+$ (ref. 32). These peaks vanished for the cells with Co, TPAA and TPAA/Co electrolyte, which indicates an efficient dye regeneration of the oxidized LEG4$^+$ in all these three electrolytes[33,34].

The bleach peak at ~600 nm was recently shown to be caused by a Stark effect on the dye ground-state absorption band, as a result of the change in electric field at TiO$_2$/dye/electrolyte interface after electron injection[32,35]. Significantly, in the case for the TPAA electrolyte, a new positive absorption peak at 720 nm

---

**Table 1 | Photovoltaic performance of DSSCs based on the LEG4 dye and co-sensitizer with varied electrolytes and varied light intensity (AM 1.5 G spectral distribution and aperture area 0.25 cm$^2$).**

| Sensitizers | Electrolyte | Light intensity (mW cm$^{-2}$) | $V_{OC}$ (mV) | $J_{SC}$ (mA cm$^{-2}$) | FF (%) | $\eta$ (%) |
|---|---|---|---|---|---|---|
| LEG4 | Co | 100 | 835 | 12.1 | 71.4 | 7.2 |
| | TPAA/Co | 100 | 915 | 14.1 | 70.2 | 9.1 |
| Co-sensitized (D35/Dyenamo blue) | Co | 100 | 820 | 13.9 | 72.4 | 8.3 |
| | TPAA/Co | 100 | 920 | 15.5 | 73.3 | 10.5 |
| | TPAA/Co | 46 | 890 | 8.2 | 73.5 | 11.7 |
| | TPAA/Co | 11.4 | 825 | 1.9 | 74.9 | 10.2 |

TPAA, tris(p-anisyl)amine.

---

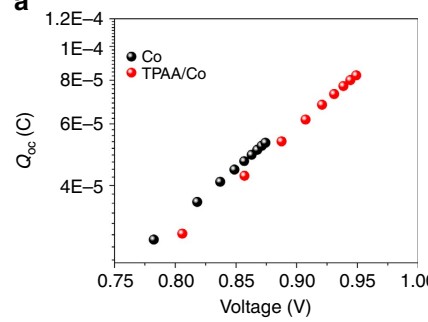

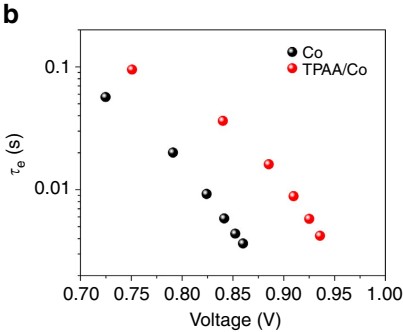

**Figure 3 | Charge extraction and electron lifetime.** Charge extraction (**a**) and electron lifetime (**b**) measurements as a function of the voltage under open circuit conditions for DSSCs sensitized with LEG4 dye employing the standard cobalt and TPAA/Co electrolyte.

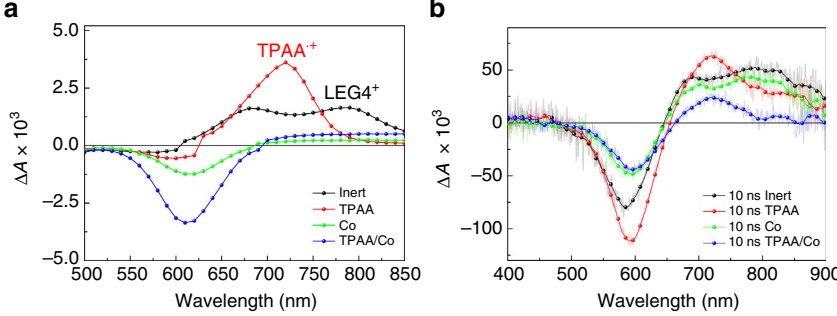

**Figure 4 | Photo-induced and transient absorption.** Photo-induced absorption spectra (**a**) and transient absorption spectra (**b**) of LEG4-sensitized TiO₂ employing four different electrolytes: inert (0.1 M LiClO₄, 0.2 M 4-tert butylpyridine (TBP)), TPAA (0.1 M TPAA, 0.1 M LiClO₄, 0.2 M TBP), Co (0.1 M LiClO₄, 0.2 M TBP, 0.22 M Co(bpy)₃(PF₆)₂ and 0.05 M Co(bpy)₃(PF₆)₃) and TPAA/Co (0.1 M TPAA, 0.1 M LiClO₄, 0.2 M TBP, 0.22 M Co(bpy)₃(PF₆)₂, and 0.05 M Co(bpy)₃(PF₆)₃ ). The transient absorption spectra were acquired after ∼10 ns following the excitation of a 600 nm laser pulse with an intensity ∼0.15 mJ to 0.25 mJ cm⁻².

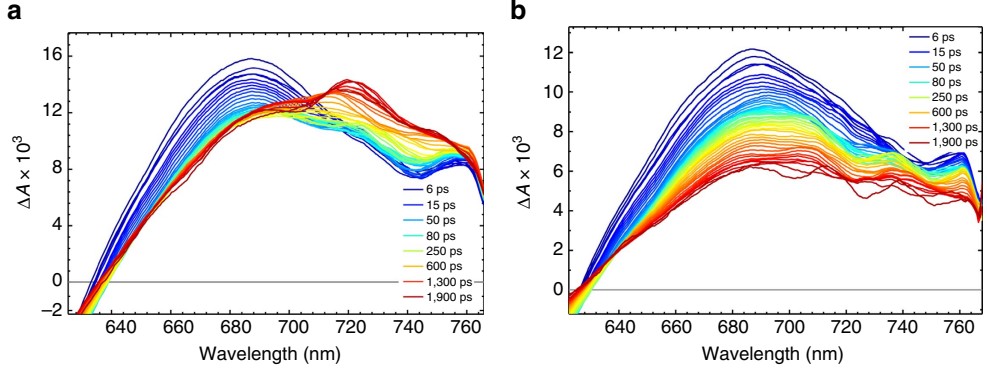

**Figure 5 | Femtosecond transient absorption spectra.** (**a**) TPAA/Co electrolyte and (**b**) Co electrolyte on LEG4 sensitized TiO₂ films. The excitation wavelength is 525 nm with an intensity of 500 nJ per pulse.

appears, which is consistent with the characteristic absorption of TPAA·⁺, as shown above by the spectroelectrochemistry measurements. The presence of TPAA·⁺ results in an efficient dye regeneration process of LEG4⁺ by TPAA. On the other hand, the absence of peak at 720 nm in the case of TPAA/Co electrolyte indicates an electron transfer process from Co(bpy)₃²⁺ to TPAA·⁺. These steps are summarized as below:

$$LEG4^+ + TPAA \rightarrow LEG4 + TPAA^{\cdot +} \quad (1)$$

$$TPAA^{\cdot +} + Co(bpy)_3^{2+} \rightarrow TPAA + Co(bpy)_3^{3+} \quad (2)$$

In the presence of the redox electrolytes, the broad and featureless PIA signal at wavelengths higher than 700 nm is attributed to electrons in mesoporous TiO₂. It is evident that the electron concentration is much higher in case of the TPAA/Co electrolyte by a factor of 2, which gives further evidence for the longer lifetime of electrons in this system, resulting in more electron accumulation. This is in excellent agreement with the electron lifetime measurements in Fig. 3.

Nanosecond laser measurements were further used to verify these two electron transfer processes. Figure 4b shows the transient absorption spectra of the DSSCs with four different electrolytes recorded 10 ns after excitation at 600 nm, which is close to the limit of the time resolution of the instrument. The transient absorption spectra spectra of both inert and Co electrolyte based DSSCs exhibit absorption peaks of LEG4⁺ at ∼ 670 and 770 nm after 10 ns. This can be understood from the fact that after the ps-fs electron injection from the excited dye to the conduction band of TiO₂, the oxidized LEG4 remains because

for some time since the regeneration by the Co electrolyte is relative sluggish, (μs scale)[21]. In the contrast, the peak of TPAA·⁺ located at 720 nm immediately appears after 10 ns after excitation for TPAA and TPAA/Co electrolyte-based DSSCs. The appearance of TPAA·⁺ instead of LEG4⁺ spectrum in the latter case indicates that an efficient regeneration of LEG4⁺ by TPAA occurs faster than 10 ns.

The kinetics of the absorbance difference, Δ*A*, at 750 nm were therefore utilized to evaluate the time scale of different electron transfer process (shown in Supplementary Fig. 3 and Supplementary Table 2). In the case of inert electrolyte, the decay process manifests the recombination of oxidized LEG4⁺ with the electrons in the TiO₂. For the case of Co electrolyte, the decay is dominated by the regeneration of LEG4⁺ by the redox mediator, [Co(bpy)₃]²⁺. In the case of TPAA and TPAA/Co electrolytes, the decay curves mainly indicate the recombination and reduction of TPAA·⁺ with the TiO₂(e⁻) and [Co(bpy)₃]³⁺, respectively, since the LEG4⁺ have already been effectively regenerated by the TPAA within in 10 ns.

Therefore, the fast regeneration process of LEG4⁺ by TPAA was also investigated with the femtosecond laser measurements and shown in Fig. 5. The initial absorption band centred ∼ 685 nm is due to LEG4⁺, in agreement with the PIA results for inert electrolyte (Fig. 5a). In the case of TPAA/Co electrolyte instead, a new absorption band at ∼730 nm grows in ∼100–1,000 ps after the laser excitation, which matches the 730 nm band of TPAA·⁺ (Fig. 5b). The combined results from both the nanosecond and femtosecond laser measurements therefore allows us to conclude that the dye regeneration process

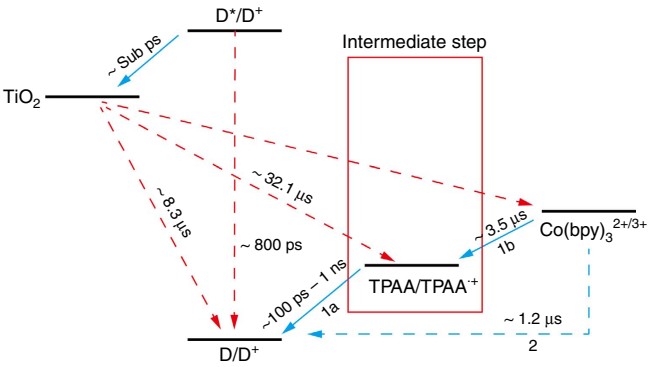

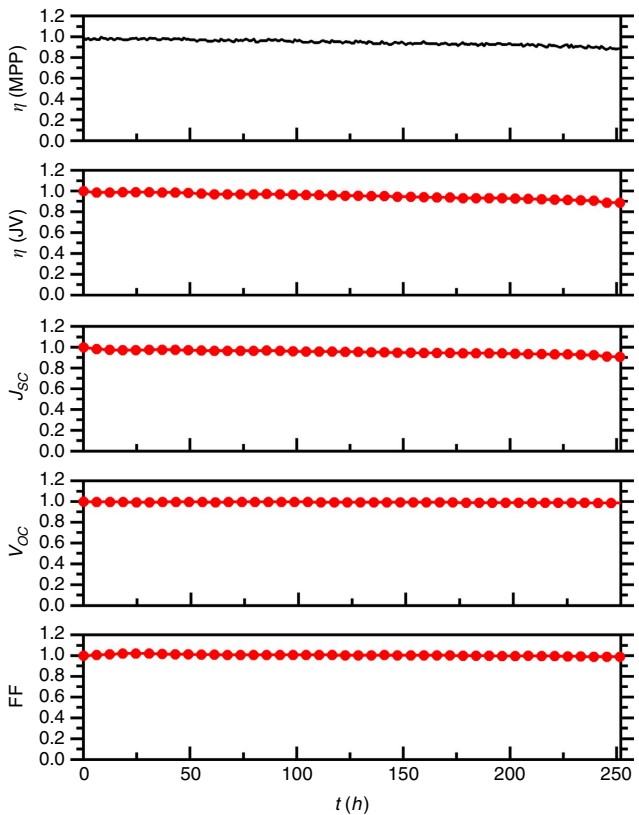

**Figure 6 | Overall charge transfer scheme.** Charge transfer schemes for the DSSCs fabricated with TPAA/Co electrolyte (path 1) and Co electrolyte (path 2). Regeneration and recombination time constants were summarized from both nanosecond and femtosecond measurements. The injection time was estimated from the fs measurement shown in Supplementary Fig. 4 and Supplementary Table 3.

between LEG4$^+$ and TPAA is ultrafast and occurs in the time interval ∼100–1,000 ps. The fast decay process (<100 ps) of the transient 685 nm band for both Co and TPAA/Co electrolytes is attributed to the ultrafast photophysical relaxation processes of the LEG4 dye and is briefly discussed in the supporting information (Supplementary Fig. 4; Supplementary Table 3).

To sum up the information mentioned above, the entire charge transfer scheme with all the information of electron transfer dynamics are plotted in Fig. 6, with emphasis on the electron transfer cascade from the [Co(bpy)$_3$]$^{2+}$ to LEG4$^+$ for the TPAA/Co electrolyte.

In these two-step electron transfer processes, the electrons firstly flow from TPAA to the oxidized dye (1a) in an ultrafast process (∼100–1,000 ps), and are then transferred between the oxidized TPAA$^{·+}$ and [Co(bpy)$_3$]$^{2+}$ (1b). Compared with the electron transfer processes of the Co electrolyte, TPAA acts as the intermediate state in the entire charge transfer processes. Consequently, this ultrafast dye regeneration by the intermediate state ensures a rapid regeneration of the oxidized dye in the present system. Previous studies by Gerald Meyer[36] and Durrant and co-workers[37] have clearly verified the importance of the recombination between TiO$_2$(e$^-$) and the oxidized dye. As a result, rapid regeneration of LEG4$^+$ by TPAA, which have the characteristic time constants of 100–1,000 ps, therefore reduces recombination process of TiO$_2$(e$^-$) and LEG4$^+$ with the slower time constant of 8.3 μs (Supplementary Table 2). Significant improvement of the device performance was thereby achieved in terms of both $V_{OC}$ and $J_{SC}$. This implies that under standard DSSC operation conditions considerable electron in TiO$_2$ to oxidized dye losses occur in the cobalt complex electrolyte system. The electron lifetime in the DSSC is increased by addition of TPAA because it removes these losses. It is interesting to notice that the similarity between the current tandem redox system with the traditional I$^-$/I$_3^-$ electrolyte, which involves intermediate states during the dye regeneration processes that ensure both a fast dye regeneration and long electron lifetime[5].

**Stability measurements**. Finally, we investigated the stability of our optimized devices for 250 h under continuous 1 Sun illumination, MPP and 25 °C. Figure 7 shows the normalized η evolution for a device with the champion cell configuration (Co-sensitizer and TPAA/Co electrolyte). Impressively, after 250 h, the best cells retained 89% of the initial efficiency. Identical η values were obtained from MPP tracking and J–Vs, showing that

**Figure 7 | Evolution of normalized photovoltaic parameters.** The photovoltaics parameters were measured over 250 h under continuous 1 sun illumination, MPP tracking and 25 °C. η (MPP; top, black line). η (J-V) and other parameters (red line and circles). MPP was tracked hourly, and J-Vs were scanned every 6 h. The initial solar cell value is: $V_{OC}$: 910 mV, $J_{SC}$: 15.1 mA cm$^{-2}$, FF (%): 70.5, η (%): 9.7.

the devices displayed negligible hysteresis. The $J_{SC}$ slightly decreases, and mirrors the overall change in efficiency. The fill factor (FF) and $V_{OC}$ remained completely stable. These results are comparable to, if not better than, previously reported stability data for cobalt electrolyte DSSCs[38]. Our study is, to the best of our knowledge, the first to demonstrate a high stability in DSSCs that use radical redox species in the electrolyte. The current density at MPP was constant at about 10 mA cm$^{-2}$ during stability test, from which is calculated that 9,000 C charge passed through the system per cm$^2$. With a dye coverage of about 10$^{-7}$ mol cm$^{-2}$, this corresponds to ∼10$^6$ turnovers for each dye molecule in the test (about one excitation and electron injection for each dye molecule per second). With ∼3 μl of electrolyte per cm$^2$, it is calculated that each TPAA molecule undergoes 3 × 10$^5$ turnovers, without any apparent degradation. This demonstrates that the use of radical organic donors (such as TPAA/TPAA$^{·+}$) in the DSSC electrolyte can be robust and viable. The volatile ACN-based electrolyte is not suited for long-term stability in DSSC. Initial measurements show that TPAA is compatible with other, more suitable solvents for long-term stability.

## Discussion

We have demonstrated a new strategy that significantly improves the performance of DSSCs via simple addition of an organic electron donor into the standard cobalt complex electrolyte. The DSSCs developed here achieved 11.7% power conversion efficiency at 0.46 sun and 10.5% at 1 sun illumination, with excellent stability. The improvement of the photovoltaic performance is attributed to the increased $V_{OC}$ and $J_{SC}$ attributed to

avoidance of recombination losses between electrons in $TiO_2$ and oxidized dye molecules. Time-resolved spectroscopic investigations directly show the change of the charge transfer processes at $TiO_2$/dye/electrolyte interface induced by the presence of TPAA, i.e., a distinguished two-step electron transfer occurring consecutively between the oxidized dye/TPAA and TPAA$^{\cdot +}$/$Co^{2+}$, with a characteristic time of $\sim$100–1,000 ps and 14 $\mu$s, respectively. We have shown that by the rapid regeneration of $D^+$ by the TPPA in the electrolyte, one path for electron recombination is reduced: the recombination of oxidized dye by electrons in the $TiO_2$. These results demonstrate new opportunities for further improvement of the DSSCs performance, but could also be of inspiration in general for photoelectrochemical systems aiming for optimized and controlled electron transfer processes.

## Methods

**Fabrication of solar cells.** $TiO_2$ photo-electrodes were prepared on the fluorine-doped tin oxide (FTO) glass, which initially was cleaned in an ultrasonic bath with detergent, water, acetone and ethanol for 30 min, respectively. Then the FTO glass was pretreated with $TiCl_4$ (40 mM in water) at 70 °C for 90 min, washed with water and dried. A screen printing technique was used to prepare mesoporous $TiO_2$ films with an area of $5 \times 5$ mm$^2$. The film consists of one transparent layer (4 $\mu$m) that was printed with colloidal $TiO_2$ paste (Dyesol DSL 30 NRD-T) and one light-scattering layer (4 $\mu$m) prepared by another paste (PST-400C, JGC Catalysts and Chemical Ltd). Before printing the second layer the film was dried at 125 °C for 6 min. Afterwards the electrodes were sintered in an oven (Nabertherm Controller P320) in an air atmosphere using a temperature gradient program with four levels at 180 °C (15 min), 320 °C (15 min), 390 °C (15 min) and 500 °C (30 min). Before the dye-sensitization the electrodes were post treated with $TiCl_4$ as mentioned above, followed by heating at 500 °C for 30 min. At a temperature of 90 °C, the electrodes were immersed in a dye bath for 18 h containing either LEG4 (0.2 mM) or mixture of D35 and Dyenamo Blue in the ratio of 4:3 (0.1 mM: 0.075 mM) in tert-butanol: ACN (1:1 (v/v)). Non-attached dyes were removed with tert-butanol: ACN (1:1 (v/v)). Counter electrodes were prepared by depositing 10 $\mu$l of a $H_2PtCl_6$ solution in ethanol (5 mM) to FTO glass substrates followed by heating in air at 400 °C for 30 min. Solar cells were assembled by sandwiching the photoelectrode and the counter electrode using a 25-$\mu$m thick thermoplastic Surlyn frame. An electrolyte solution was then injected through a hole predrilled in the counter electrode by vacuum back filling and the cell was sealed with thermoplastic Surlyn cover and a microscope glass coverslip. The Standard Co electrolyte consists of $Co(bpy)_3(PF_6)_2$ (0.22 M), $Co(bpy)_3(PF_6)_3$ (0.05 M), $LiClO_4$ (0.1 M) and 4-tert butylpyridine (0.2 M) in ACN and TPAA/Co electrolyte is made by adding 0.1 M TPAA into standard Co electrolyte.

**Solar cells characterization.** Current–voltage (IV) characteristics were determined by using a combination of a source measurement unit (Keithley 2400) and a solar simulator (Newport, model 91160). The solar simulator was giving light with AM 1.5 G spectral distributions and was calibrated to an intensity of 100 mW cm$^{-2}$ using a certified reference solar cell (Fraunhofer ISE). On top of the DSC a black metal mask with an aperture of $5 \times 5$ mm$^2$ was applied.

Incident IPCE spectra were measured with a computer-controlled set-up comprising a xenon light source (Spectral Products ASB-XE-175), a monochromator (Spectral Products CM110) and a Keithley multimeter (model 2,700). The IPCE spectra were calibrated using a certified reference solar cell (Fraunhofer ISE).

Electron lifetime as a function of extracted charge at different bias light intensities was investigated in a toolbox set-up as described previously[7,31]. A white light-emitting diode (LED; Luxeon Star 1W) was used as a light source. Transient photo-voltage response of the DSSCs was recorded using a 16-bit resolution digital acquisition board (National Instruments) in combination with a current amplifier (Stanford Research Systems RS570) and a custom electromagnetic switching system. The transient photo-voltage was recorded by overlapping the bias light with a small square-wave modulation and the response was subsequently fitted to a first-order exponential function. For the charge extraction measurements the DSSCs were illuminated for 5 s at the same bias light intensities as for the electron lifetime measurements. After 5 s the LED is turned off, the external circuit is short-circuited and the current density is read and integrated over time.

Device stability was measured using continuous 1 Sun equivalent illumination via white LEDs, which were calibrated using a Si photodiode; the illumination intensity was calibrated to produce the same photocurrent as under AM 1.5 solar simulator illumination. The MPP was tracked and maintained, using an in-house built MPP-tracking robot. Temperature was maintained at 25 °C via fans and water-cooling. MPP voltage was tracked and applied, hourly. $J$–$V$s were measured every 6 h, and scanned cyclically (10 mV voltage step, 0.1 V S$^{-1}$ scan rate) from MPP voltage to $+1$ V, $-0.1$ V and return.

**Electrochemical characterization.** A three-electrode cell was used for the cyclic voltammetry experiments by using Autolab Potenstat 10. The supporting electrolyte used for electrochemical measurements was 0.1 M TBAPF$_6$ in ACN. The counter electrode is graphite. The reference electrode was Ag/AgCl with 1 M LiCl in ethanol. The working electrode is gold. The reference electrode was calibrated by recording the cyclic voltammogram of ferrocene in the same electrolyte; the potential values are on the basis of the estimated value of the ferrocene redox potential in ACN 0.624 V with respect to NHE[39].

**Photo-induced spectroscopy.** PIA measurements were performed using the same set-up as described in previous literature[34]. A square-wave modulated blue LED (Luxeon Star 1 W, Royal Blue, 460 nm) was used for excitation, superimposed on a continuous white probe light provided by a 20 W tungsten–halogen lamp. The transmitted probe light was focused onto a monochromator (Acton Research Corp. SP-150), detected using a ultraviolet-enhanced Si photodiode and connected to a lock-in amplifier (Stanford Research Systems model SR830) via a current amplifier (Stanford Research Systems model SR570). The intensity of the probe light was $\sim$20 mW cm$^{-2}$, and the intensity of the excitation LED was $\sim$8 mW cm$^{-2}$. The modulation frequency of the LED was 9.33 Hz.

**Nanosecond laser spectroscopy.** Samples are prepared according to previous procedures[40]. Basically, transparent $TiO_2$ films were sensitized to act as the photocathode, while non-platinized FTOs were used as the counter electrode. The sensitized films with LEG4 were used in present study, all of which assembled with four different electrolytes (the same electrolyte as described in the PIA figure caption).

Laser measurements were performed with a laser flash photolysis spectrometer (Edinburgh Instrument LP920), using a continuous wave xenon light as a probe light. Laser pulses were generated and tuned to 600 nm, using a frequency tripled Nd:YAG laser (Continuum Surelight II, 10 Hz repetition rate, 10 ns pulse width) in combination with an OPO (Continuum Surelight). The pulse intensity was attenuated to 0.15–0.25 mJ cm$^{-2}$ per pulse using neutral density filters, so that only few electrons are injected into every $TiO_2$ nanoparticle on the pulse excitation of dye molecules. The full transient absorption spectrum was recorded with a charge-coupled device camera. The gate delay and gate width was set to be 10 ns, close to the limitation of the instruments. Kinetic traces were monitored at 750 nm, where oxidized LEG4 shows strong absorption indicated by PIA measurement (see Supporting information). The signal was averaged from 100 to 500 pulses to reduce the noise.

**Ultrafast laser spectroscopy.** The preparation of LEG4-sensitized films ($ZrO_2$ or $TiO_2$) for the femtosecond transient absorption measurements were similar to those in the nanosecond measurements. The electrolytic composition are shown above, except for that propylene carbonate was used as the solvent to avoid the volatility of ACN. During the measurements, a drop of electrolyte solution was added to the sensitized film and covered with a microscopic glass by capillary force. The absorbance of the sample films is controlled between 0.3–0.8.

Femtosecond time-resolved measurements were done in the mode of transient absorption. A detailed description is refer to Petersson et al.[41] Briefly, the output from a Coherent Legend Ti:Sapphire amplifier (1 kHz, $\lambda = 800$ nm, FWHM (full-width at half-maximum) 100 fs) was split into a pump and a probe part. Desired pump wavelengths were obtained with a non-collinear optical parametric amplifier (TOPAS Prime). The energy of each pulse was controlled between 150 and 500 nJ over ca. 3 mm$^2$ by use of the natural density filter. The white light continuum was used as the probe light and obtained by focusing part of the 800 nm light on a moving CaF$_2$ plate. Polarization of the pump was set at magic angle, 54.7°, relative to the probe. Instrumental response time depends on pump and probe wavelengths, but is typically $\sim$150 fs.

Data analysis are done in MATLAB (The MathWorks, Inc.), a robust trust-region reflective Newton nonlinear-least-squares method is used for the fits of time traces. Traces ($\Delta A$ vst) are fitted to a sum of exponentials convolved with a Gaussian-shaped response. Also included in the fits is an artifact signal that is due to the cross phase modulation during pump and probe overlap. All spectra are corrected for chirp in the white light probe. The time zero is set at maximum pump-probe temporal overlap.

**Data availability.** The data that support the findings of this study are available from the authors upon request.

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

## Acknowledgements

We gratefully acknowledge the Swedish Energy Agency, the Swedish Research Council (VR), the STandUP for Energy program, the Knut and Alice Wallenberg and Stiftelsen Olle Engkvist Byggmästare Foundation for the financial support. W.Y. and L.Z. acknowledge the China Scholarship Council (CSC) for a PhD study fellowship. Dr Brian O'Regan is acknowledged for contributing to the design and programming of the MPP-tracking robot. Kari Sveinbjörnsson is acknowledged for purifying TPAA compounds and Rodrigo Garcia is acknowledged for preparing counter electrodes.

## Author contributions

Y.H. designed, fabricated and optimized the solar cells, with help from Y.S and W.Y. And then Y.H. conducted the IV, IPCE and electron lifetime, charge extraction measurements. W.Y. and Y.H conducted the PIA and nanosecond laser experiments. L.Z. and W.Y. designed and carried out the femtosecond laser measurements together. W.Y. and E.M. measured the electrochemical properties. R.J. conducted the stability measurements. Y.H. and W.Y. wrote the manuscript together, with contribution from all authors. L.H. and A.H. have given technical support and scientific feedbacks during the whole project. G.B. supervised the whole project.

## Additional information

**Competing financial interests:** The authors declare no competing financial interests.

**Publisher's note**: 

