## [Peer Review File · Nature Communications]

Reviewer #1 (Remarks to the Author)

The authors reported application of a small electron donor in cobalt complex electrolyte for efficient improve the efficiency of dye-sensitized solar cells and with high long-term stability. The research result is important for preparing high performance and stable DSSCs with cobalt based electrolyte. It is novel and valuable. After minor revision, the paper can be suggested to published in the journal. Namely, some important related references should be added such as Jihuai Wu, Zhang Lan, Jianming Lin, Miaoliang Huang, Yunfang Huang, Leqing Fan, Genggeng Luo, Electrolytes in dye-sensitized solar cells, Chemical Reviews, 2015, 115(5), 2136-2173; Zhang Lan, Jihuai Wu, Jianming Lin, Miaoliang Huang, A highly efficient electric additive for enhancing photovoltaic performance of dye-sensitized solar cells, Science china chemistry, June 2010 Vol.53 No.6: 1352-1357.

Reviewer #2 (Remarks to the Author)

Comments on 'A Small Electron Donor in Cobalt Complex Electrolyte Significantly Improves Efficiency in Dye-Sensitized Solar Cells'.

This is a quite interesting work, showing an increase of the quantum efficiency for DSSCs, simply by adding organic electron donor, tris(4-methoxyphenyl)amine (TPAA), into the cobalt tris(bipyridine) electrolyte.

Power conversion efficiency of 11.7 % at 0.46 sun illumination and 10.5 % efficiency at one sun could be achieved, which can attributed to the intermediate function of the organic redox mediator enabling fast and efficient regeneration of the oxidized dye and effectively suppressing electron recombination from TiO₂ to the oxidized dye.

This article is well organized, which deserves publication.

1. The authors show the normalized curves of PCE, JSC, VOC, FF. The specific number of cells tested and average values with error should be recorded in the manuscript to ensure the stability and repeatability of this work. In addition, a new electrolyte must be used for the stability testing for the solvent of acetonitrile would be helpful for this.
2. The highest PCE is only 10.5% in this work, which is poor as compared to other similar works. I would suggest to use more efficient sensitizers to demonstrate their strategy.
3. The DSSC actually is a complicated electrochemical system already. The authors introduce the concept of 'tandem electrolyte', which add the complex for this. This is not in a good logic for the multi-step electron transfer would slow down the reaction. This point should be clearly discussed in the revised ms.
4. the device uses TPAA electrolyte should be provided
5. for the devices with co-electrolytes, they showed higher output voltage. The authors ascribe this to the ultrafast dye regeneration kinetics, and slowed down recombination between the electrons in TiO₂ and both oxidized dye and oxidized redox species. Therefore, a longer electron lifetime was given to prove this statement. This is not enough to give a clear conclusion. More evidence should be given.
6. the stability testing was carried out with LED light sources. Why? The standard test condition would be using Xe lamp. This is important for the TPAA+ showed photo-activity at around 720 nm.

Reply to Reviewers' comments:

We thank the reviewers for their insightful comments. We have reproduced their feedback below, our reply is printed in **bold red font**.

Reviewer #1 (Remarks to the Author):

The authors reported application of a small electron donor in cobalt complex electrolyte for efficient improve the efficiency of dye-sensitized solar cells and with high long-term stability. The research result is important for preparing high performance and stable DSSCs with cobalt based electrolyte. It is novel and valuable. After minor revision, the paper can be suggested to published in the journal. Namely, some important related references should be added such as Jihuai Wu, Zhang Lan, Jianming Lin, Miaoliang Huang, Yunfang Huang, Leqing Fan, Genggeng Luo, Electrolytes in dye-sensitized solar cells, Chemical Reviews, 2015, 115(5), 2136-2173; Zhang Lan, Jihuai Wu, Jianming Lin, Miaoliang Huang, A highly efficient electric additive for enhancing photovoltaic performance of dye-sensitized solar cells, Science china chemistry, June 2010 Vol.53 No.6: 1352-1357.

Reply: We thank the reviewer for favorable comments. In the revised version we have included the suggested review paper, but not the article, since its topic was not directly related to this work, in our opinion.

Reviewer #2 (Remarks to the Author):

Comments on 'A Small Electron Donor in Cobalt Complex Electrolyte Significantly Improves Efficiency in Dye-Sensitized Solar Cells'.

This is a quite interesting work, showing an increase of the quantum efficiency for DSSCs, simply by adding organic electron donor, tris(4-methoxyphenyl)amine (TPAA), into the cobalt tris(bipyridine) electrolyte.

Power conversion efficiency of 11.7 % at 0.46 sun illumination and 10.5 % efficiency at one sun could be achieved, which can be attributed to the intermediate function of the organic redox mediator enabling fast and efficient regeneration of the oxidized dye and effectively suppressing electron recombination from TiO₂ to the oxidized dye.

This article is well organized, which deserves publication.

1. The authors show the normalized curves of PCE, JSC, VOC, FF. The specific number of cells tested and average values with error should be recorded in the manuscript to ensure the stability and repeatability of this work. In addition, a new electrolyte must be used for the stability testing for the solvent of acetonitrile would be helpful for this.

Reply: Statistics are indeed important when assessing the solar cell performance. We have therefore added in the supporting information performance of a large series of solar cells with and without TPAA added in the electrolytes, see Supplementary Fig. 6 . We discuss these results in the main manuscript. In the long term study we have now included the original solar cell parameters in the caption of Fig.7 for completeness. This data was for one typical well-performing solar cell. Since our lab devices are not specially sealed for long-term stability test and contain acetonitrile, not all devices show good stability due to leakage, etc. The presented results demonstrate that the chemistry of the new redox system is stable. Testing of more long-term stable electrolyte solvent is not part of this study. This will be done in future research.

2. The highest PCE is only 10.5% in this work, which is poor as compared to other similar works. I would suggest to use more efficient sensitizers to demonstrate their strategy.

Reply: No anti-reflecting film was used in present work, which could increase efficiency by at least 5% relative. High PCEs of 12 % and more are only obtained in a few laboratories and can generally not be reproduced elsewhere; In fact, the certified DSSC efficiency lies at 11.9%, which is close to the results obtained here. We are very confident that other groups can reproduce our results.

We did some initial tests with a porphyrin dye named YD2-o-C8.^[1] However, no very efficient solar cells were obtained. We agree with the reviewer that better dyes should improve PCE significantly. However, due to the limitation of dyes in our lab, this is the best sensitizer

system we have so far. It is the new redox system and its mechanism insight that is the focus of the present study. Further design and synthesis of more efficient dye is of great interest, but deserves a separate study.

We added to this point in the revision:

It should be noted that the investigated devices did not have an anti-reflecting coating. Further optimization of mesoporous TiO₂, dye layer and electrolyte layer will likely increase PCE of this system even further.

3. The DSSC actually is a complicated electrochemical system already. The authors introduce the concept of 'tandem electrolyte', which adds the complexity for this. This is not in a good logic for the multi-step electron transfer would slow down the reaction. This point should be clearly discussed in the revised ms.

Reply: We provide a clear explanation why fast regeneration by TPAA addition improves the solar cell, even though admittedly it complicates the overall mechanism. We provide a complete overview of its function and the reason to use TPAA, see for example Fig. 6.

The result of TPAA addition to cobalt electrolyte is that recombination between electrons in the TiO₂ nanoparticles (TiO₂(e⁻)) with the oxidized dye, LEG4⁺ is largely suppressed, which improves device performance.

TPAA itself is a simple chemical and does not add significantly to the device cost.

4. the device uses TPAA electrolyte should be provided

Reply: We have added the result for DSSCs with pure TPAA redox electrolyte in supporting information and discuss it briefly in the main manuscript. Due to fast electron recombination to TPAA^{•+}, DSSC performance is rather low. In the revised manuscript we write:

For completeness, we also tested devices with TPAA / TPAA^{•+} as the redox couple in the electrolyte. These solar cells worked, but had rather poor performance with a PCE of about 2%, see Supplementary Fig. 7 in supporting information. The relatively poor performance is attributed to rapid electron recombination from TiO₂ to the TPAA^{•+} radical cation.

5. for the devices with co-electrolytes, they showed higher output voltage. The authors ascribe

this to the ultrafast dye regeneration kinetics, and slowed down recombination between the electrons in TiO₂ and both oxidized dye and oxidized redox species. Therefore, a longer electron lifetime was given to prove this statement. This is not enough to give a clear conclusion. More evidence should be given.

Reply: Electron lifetime measurements are established and reliable measurements for the electron recombination process, sufficient to draw solid conclusions. Further evidence comes from photoinduced absorption measurements, this is added to the revised manuscript:

In the presence of the redox electrolytes, the broad and featureless PIA signal at wavelengths higher than 700 nm is attributed to electrons in mesoporous TiO₂. It is evident that the electron concentration is much higher in case of the TPAA/Co electrolyte by a factor of 2, which gives further evidence for the longer lifetime of electrons in this system, resulting in more electron accumulation. This is in excellent agreement with the electron lifetime measurements in Fig. 3.

6. the stability testing was carried out with LED light sources. Why? The standard test condition would be using Xe lamp. This is important for the TPAA⁺ showed photo-activity at around 720 nm.

Reply: White LED light sources offer the prospect of illumination without excessive heating. The sole purpose of this test was to assess the chemical stability of DSSC system under normal operation. We have therefore added this part in the main text:

The current density at MPP was constant at about 10 mA cm⁻² during stability test, from which is calculated that 9000 C charge passed through the system per cm². With a dye coverage of about 10⁻⁷ mol cm⁻², this corresponds to about 10⁶ turnovers for each dye molecule in the test (about one excitation and electron injection for each dye molecule per second). With about 3 microliter of electrolyte per cm², it is calculated that each TPAA molecule undergoes 3×10⁵ turnovers, without any apparent degradation.

since we used ACN and a simple sealing method, no true long-time stability can be expected due to leakage, etc.

The white LED (4000 K color temperature) provides excitation of the TPAA⁺ radical (see the 4000 K LED spectrum below), although it is at relatively low intensity. Nevertheless, we have no reason to believe that excitation of TPAA⁺ leads to rapid degradation.

Figure 12. Color spectrum of LXML-PW51 emitter, integrated measurement.

- [1] A. Yella, H.-W. Lee, H. N. Tsao, C. Yi, A. K. Chandiran, M. K. Nazeeruddin, E. W.-G. Diao, C.-Y. Yeh, S. M. Zakeeruddin, M. Grätzel, *Science* **2011**, 334, 629-634.

Reviewer #1 (Remarks to the Author)

Although the authors did good work by adding a small electron donor in the electrolyte to greatly enhance the performance of dye-sensitized solar cells. The structure of the dye-sensitized solar cells are still liquid-electrolyte based, especially with the highly volatile acetonitrile solvent. The work do not substantially solve the existed big problem of dye-sensitized solar cells. So its contribution is not so important for the development of the field nowadays. Owing to these reasons, it is hard for me to suggest to accept the paper to be published in the journal.

Reply to Reviewers' comments:

We thank the reviewer for the comment. We have reproduced the feedback below, our reply is printed in **bold red font**.

Reviewers' comments:

Reviewer #1 (Remarks to the Author):

Although the authors did good work by adding a small electron donor in the electrolyte to greatly enhance the performance of dye-sensitized solar cells. The structure of the dye-sensitized solar cells are still liquid-electrolyte based, especially with the highly volatile acetonitrile solvent. The work do not substantially solve the existed big problem of dye-sensitized solar cells. So its contribution is not so important for the development of the field nowadays. Owing to these reasons, it is hard for me to suggest to accept the paper to be published in the journal.

Reply: We do not claim to solve volatility problems of dye-sensitized solar cells, but provide a pathway for more efficient cells. This pathway can also be used in more stable liquid electrolyte DSSC and in solid-state DSSC. This will be part of our future work.